# ROYAL SOCIETY
# OPEN SCIENCE

### Subject Areas:
health and disease and epidemiology/differential equations/mathematical modelling

### Keywords:
lymphatic filariasis, mathematical modelling, sensitivity analysis, parameter estimation, optimal control, Caraga Region, the Philippines

### Author for correspondence:
Pamela Kim N. Salonga
e-mails: pksalonga@math.upd.edu.ph,
pnsalonga@up.edu.ph

# A mathematical model of the dynamics of lymphatic filariasis in Caraga Region, the Philippines

Pamela Kim N. Salonga[1,2], Victoria May P. Mendoza[1,2], Renier G. Mendoza[1,2] and Vicente Y. Belizario Jr[3]

[1]Institute of Mathematics, and [2]Natural Sciences Research Institute, University of the Philippines Diliman, Quezon City, Philippines
[3]College of Public Health and Neglected Tropical Diseases Study Group, National Institutes of Health, University of the Philippines Manila, Philippines

PKNS, 0000-0002-5522-1658; VMPM, 0000-0003-0953-9822; RGM, 0000-0003-3507-0327

Despite being one of the first countries to implement mass drug administration (MDA) for elimination of lymphatic filariasis (LF) in 2001 after a pilot study in 2000, the Philippines is yet to eliminate the disease as a public health problem with 6 out of the 46 endemic provinces still implementing MDA for LF as of 2018. In this work, we propose a mathematical model of the transmission dynamics of LF in the Philippines and a control strategy for its elimination using MDA. Sensitivity analysis using the Latin hypercube sampling and partial rank correlation coefficient methods suggests that the infected human population is most sensitive to the treatment parameters. Using the available LF data in Caraga Region from the Philippine Department of Health, we estimate the treatment rates $r_1$ and $r_2$ using the least-squares parameter estimation technique. Parameter bootstrapping showed small variability in the parameter estimates. Finally, we apply optimal control theory with the objective of minimizing the infected human population and the corresponding implementation cost of MDA, using the treatment coverage $\gamma$ as the control parameter. Simulation results highlight the importance of maintaining a high MDA coverage per year to effectively minimize the infected population by the year 2030.

## 1. Introduction

As a tropical country, the Philippines is endemic to several neglected tropical diseases (NTDs) such as lymphatic filariasis (LF), schistosomiasis, soil-transmitted helminths, food-borne

trematodiases, rabies and leprosy [1]. These diseases are known to affect the most neglected fraction of society—the people living in poverty in remote areas with little to no access to clean water, quality education and proper sanitation [2].

Although no individual NTD demands a global priority in terms of burden of disease due to disability and mortality, collectively, the disability-adjusted life years associated with NTDs is tantamount to that of HIV/AIDS, tuberculosis or malaria [3]. At present, NTDs threaten more than one billion people in tropical and subtropical countries worldwide [2].

LF is a parasitic disease caused by the filarial roundworms *Wuchereria bancrofti*, *Brugia malayi* and *Brugia timori* which are transmitted to humans by mosquitoes of the genera *Aedes*, *Anopheles*, *Culex* and *Mansonia*. These parasites are known to target the human lymphatic system causing different degrees of clinical signs and symptoms. While majority of the infected population are asymptomatic, many develop acute clinical disease that commonly manifests as episodic occurrence of painful inflammation of the lymph nodes and lymphatic vessels. These episodes are usually accompanied by fever, malaise, chills and headache, and are a sign of the presence of immature larvae called microfilariae (Mf) in the lymphatics. Without proper care and treatment, these periodic acute manifestations can develop into chronic disease characterized by the abnormal enlargement or swelling of body parts, usually the limbs and genitalia, due to chronic lymphatic obstruction, lymph fluid accumulation, tissue swelling and later, skin thickening. This explains why LF is commonly known as elephantiasis [4]. Although not fatal, the acute and chronic manifestations of LF can significantly diminish the quality of life of affected individuals. The debilitating and disabling symptoms associated with LF reduce the economic productivity of an infected person and their caretakers which further contributes to poverty. This, together with the social stigma caused by the disfiguring manifestations of the disease, threaten the psychosocial health of affected individuals forcing them into isolation and depression [5].

In 2000, the World Health Organization (WHO) launched the Global Programme to Eliminate Lymphatic Filariasis (GPELF) as a response to the World Health Assembly's resolution to eliminate LF as a public health problem. The GPELF endorses a two-point strategy in eliminating lymphatic filariasis: interrupting transmission through mass drug administration (MDA) and controlling morbidity [6]. The objective of MDA is to reduce microfilaremia prevalence in infected individuals to levels where infection is deemed to be intransmissible by delivering a two-drug combination of antifilarial drugs (albendazole (ALB) plus diethylcarbamazine citrate (DEC) or ivermectin) once a year to eligible people in all established endemic areas for at least 5 years. If the Mf prevalence in an area reaches elimination levels, MDA is stopped. The area then goes under surveillance to monitor infection levels for several years. If low infection levels are consistently maintained for 5 years, then the area is declared free from LF [7].

The general impact of MDA in eliminating LF is heavily dependent on the treatment coverage or the proportion of the total population in an endemic area that ingested the antifilarial drugs [8]. As per WHO guidelines [9], a minimum target coverage of 65% of the total population must be achieved per round of implementation for MDA to be effective. Reports from countries that successfully eliminated the disease highlight the importance of high MDA coverage along with the commitment and dedication of the government and health workers, and the active participation of the community in achieving elimination of LF [10–14].

The WHO also acknowledges the impact of vector control strategies in interrupting disease transmission by reducing host–vector contacts, and promotes integrated vector management against many vector-borne diseases such as malaria, dengue and LF [15]. Several studies highlight the importance of the integration of vector control with MDA in enhancing the control and elimination of lymphatic filariasis especially in highly endemic areas [16–20]. Some studies have also shown that vector control alone (i.e. without MDA) can result to a significant reduction in LF infection in the community as observed in Togo [21], Solomon Islands [22,23], The Gambia [24], Kenya [25,26], Nigeria [27] and Papua New Guinea [28–30].

In the Philippines, LF was first described in 1907 by foreign workers, and had reportedly become prevalent in several Philippine provinces towards the end of the twentieth century [31]. According to the Philippine Department of Health (DOH), LF was endemic in 46 (out of 81) provinces in 12 (out of 17) regions with an at risk population of 45 000 000 Filipinos [1]. As part of the DOH's commitment to eliminate the disease as a public health problem in the country, it has created the National Filariasis Elimination Program (NFEP), previously called National Filariasis Control Program, which has adopted GPELF's elimination strategy. Since 2001, MDA using the combination drug DEC-plus-ALB is implemented once every year for a minimum of 5 years in all established endemic areas in the country. The DOH has set a target programme coverage of at least 85% annually to increase the chances of achieving Mf prevalence and antigen rate of less than 1%, which is the set criterion for stopping LF MDA in the country [1]. The NFEP also continues to strengthen the morbidity management and disability prevention strategy to alleviate the suffering and disability of chronically infected individuals.

Vector control measures such as spraying and use of bednets are generally not a supported strategy for most LF endemic areas in the country due to their underwhelming impact in reducing host–vector contacts. This inefficiency can be attributed to the exophilic, exophagic and day-biting behaviours of *Aedes poicilius* mosquitoes, which are the principal vectors of LF in the Philippines [32]. These mosquito vectors are known to breed on the leaf axils of abaca and banana plants; hence, any form of vector control that aims to destroy their breeding sites may result to the loss of livelihood of farmers and field workers in these areas [33]. However, in areas wherein *Anopheles* mosquitoes are the main vectors of LF, vector control measures can enhance the impact of MDA in interrupting the transmission [7].

As of 2018, LF threatens around 893 million people in 49 countries worldwide and a global baseline estimate of 36 million individuals are suffering from chronic disease manifestations [8]. This is a remarkable progress compared to the 1.3 billion people at risk and 120 million people infected with 40 million disfigured and incapacitated in 2000 [34]. However, the magnitude of these numbers reflect how far we are still from our goal of eliminating LF as a global public health problem. Because of this, the WHO has reset its target for elimination of LF to the year 2030 from its initial target of 2020. According to the new NTD roadmap [35], at least 58 out of 72 (81%) endemic countries need to be validated for the elimination of LF as a public health problem by 2030. Unfortunately, the Philippines is one of the countries that are yet to eliminate LF as a public health problem with approximately 5.3 million Filipinos in 6 endemic provinces still requiring MDA as of 2018, which is the DOH's national elimination target date for LF [1,36].

Three general simulation models of LF transmission and control (LYMFASIM [37], EPIFIL [38,39] and TRANSFIL [40]) are currently being used to support policy making and designing elimination strategies for LF. LYMFASIM is an individual-based modelling framework wherein different models for the transmission and control of LF can be composed by choosing different assumptions and by varying parameter values [37]. EPIFIL is a coupled partial differential equation and ordinary differential equation model describing patterns of LF infection and associated diseases through the changes in the human and mosquito populations over age and time [38]. TRANSFIL is described to be the individual-based stochastic equivalent of EPIFIL [40]. All three models account for the human, mosquito and parasite dynamics in the transmission and control of LF. Some studies on these simulation models are discussed in [41–46].

Several mathematical models of LF have also looked into the dynamics of the disease by considering the interaction of the human and mosquito populations. In 2009, Supriatna and Soewono investigated the long-term effects of targeted medical treatment in the disease dynamics in Jati Sampurna, West Java, Indonesia using an SI–SI model [47]. Supriatna extended this model by considering an additional human compartment consisting of latent individuals to account for the 'delay' in the infection period [48]. Numerical simulations using an SEI-SI model developed by Bhunu and Mushayabasa suggest that treatment of both exposed and infected humans might lead to a more effective control of the disease compared to the treatment of infected population alone [49]. This model is extended by Bhunu to assess the potential of pre-exposure vaccination and chemoprophylaxis in the control of the disease [50]. In a 2016 study, Iddi *et al.* assessed the impact of MDA, health education campaigns and the vector control sterile insect technique on the transmission dynamics of LF using a deterministic model [51]. In 2017, Mwamtobe *et al.* proposed an SEI-SI LF model with quarantine and treatment as control strategies [52]. In these studies, the parameter values are either assumed or obtained from existing malaria models.

Herein, we develop a mathematical model of LF to investigate how mass drug administration impacts the disease dynamics in the Philippines. Compared to the existing models which mostly explored scenarios wherein the treatment is given only to the infectious population, we consider a more realistic representation of MDA wherein the antifilarial drugs are given to all eligible individuals in the population, infected and uninfected alike. Hence, the treatment affects not only the dynamics of the infectious human population but also those who are infected but not yet infectious. Since this study is Philippine-specific, we also use Philippine filariasis data to estimate model parameters to have more meaningful insights. The model considers no intervention on the vector population due to the reasons mentioned previously.

Ultimately, we aim to use optimal control theory (OCT) to determine the best implementation strategy for MDA until 2030 that will lead to a reduction of the infected population with a minimized MDA implementation cost. There are various applications for OCT in many different fields. For instance, in modelling cancer dynamics, OCT can be used to determine the optimal treatment regimen which will minimize tumour density and drug side-effects over a defined period of time [53]. In epidemic modelling, OCT can be used to determine optimal vaccination schedule which will lead to the elimination of the epidemic in the population [54]. In modelling disease transmission and control, OCT

can provide insights on the optimal intervention strategy with the least implementation cost [55]. Other studies with applications of optimal control theory can be found in [56,57] (tuberculosis), [58] (dengue) and [59] (HIV/AIDS). An in-depth discussion on optimal control theory and a number of insightful applications are provided by Lenhart *et al.* [60].

To the authors' knowledge, this study provides the first mathematical model of LF in the Philippines. Although the DOH begun MDA in 2001, we still see Mf prevalence rates of greater than 1% in some areas in the Philippines. This motivates us to study the transmission dynamics of LF and suggest ways to accelerate the elimination of the disease especially in the remaining endemic areas in the country. This study aims to assist the DOH and similar programmes in the region in designing more effective and cost-efficient implementation approaches for MDA fit for each endemic area, to achieve LF elimination in the whole country in the near future.

The rest of the paper is organized as follows. In the *Methods* section, the proposed model for LF transmission is described. Information on the epidemiological data and the parameter values and initial conditions used are also presented in this section. In the *Results and discussion* section, stability analysis of the steady state solutions of the model is presented. Results of the sensitivity analysis using the Latin hypercube sampling and partial rank correlation coefficient method are also discussed. The obtained results provide information on the critical model parameters with respect to the infected population, which guide the parameter estimation using the available filariasis data from the Philippine DOH. In this section, numerical simulations on the application of optimal control theory using the forward–backward sweep method are also presented. We investigate the efforts in minimizing the number of infected individuals and the corresponding implementation cost of MDA. We summarize the results of our work and recommend possible extensions of this research in the last section.

# 2. Methods

## 2.1. Mathematical model of lymphatic filariasis

Filarial parasites that cause LF need two host species to complete their five-stage life cycle: a definitive host (humans or animals) wherein the development from third-stage larva (L3) to adult worm and the reproduction of microfilariae occurs, and an intermediate host (mosquito) wherein the development from microfilaria to L3 occurs. The mosquito also acts as a vector of the parasite that physically carries and transmits infective larvae from one human to another. Without one host or the other, disease transmission will not be sustained in the population. Mosquitoes are infectious to humans if they harbour third-stage larval parasites and humans are infectious to mosquitoes if they harbour microfilariae [61].

The LF transmission dynamics is summarized as follows. When an infected mosquito bites an uninfected human, the infective L3 are introduced onto the skin. The infection in humans begins when these parasites enter the human body through the mosquito bitewound and migrate to the lymphatics where they develop to maturity within 6–12 months [62]. Adult worms reproduce and fecund female worms release thousands of microfilariae. These microfilariae travel through the lymphatic channels into the blood stream where they are taken up by a mosquito through a blood meal. Within 10–12 days, the ingested microfilariae move from the mosquito's gut to its thoracic cavity where they mature to infective L3. At this point, the L3 larvae migrate to the mosquito's proboscis, and the transmission cycle continues when the infected mosquito bites an uninfected human [61].

We propose a deterministic model of LF transmission involving the interaction of the human and mosquito populations. First, we assume that in an LF endemic area, the human population can be categorized into three epidemiological classes based on each individual's infection level: uninfected $U_h(t)$, latent $L_h(t)$, and infectious $I_h(t)$. The latent stage accounts for the development of infective L3 larvae into fecund adult worms. Hence, individuals in the latent stage are considered infected but not yet infectious. Meanwhile, all infected individuals who are able to transmit the infection are in the infectious class. Thus, the total human population is represented by $N_h(t) = U_h(t) + L_h(t) + I_h(t)$. On the other hand, mosquitoes can only be either uninfected $U_v(t)$, or infected $I_v(t)$. Here, the latent stage is considered negligible; thus, infected vectors are assumed to be infectious. We note that the model considers only the female mosquito population since only adult female mosquitoes contribute to the transmission. Hence, the total mosquito population at time $t$ is defined as $N_v(t) = U_v(t) + I_v(t)$. Moreover, the model considers one species of worm and one species of mosquito.

We assume that recruitment into both human and mosquito populations is only by birth. Moreover, since there is no vertical transmission of the infection (i.e. infected pregnant mothers cannot pass the infection to

their offspring), all new members of both populations enter their respective uninfected classes at *per capita* rates $b_h$ and $b_v$. Since there is no disease-induced death, both humans and mosquitoes leave their respective population through natural death at respective *per capita* rates $\delta_h$ and $\delta_v$.

The model also assumes homogeneous mixing of the human and mosquito populations. That is, each uninfected individual in the population has an equal probability of being bitten by an infected mosquito and each uninfected mosquito has the same probability of biting an infected human. Now, if $\beta$ is the average number of bites per mosquito per unit time, then there are $\beta(N_v(t)/N_h(t))$ mosquito bites per human per unit time. However, only a fraction of these bites, which come from infected mosquitoes, are potentially infective to humans. Further, only a proportion of the potentially infective bites actually result to a successful transmission. Thus, we define the force of infection from mosquitoes to humans, $\lambda_{vh}(t)$, as the product of the average mosquito bites per human per time $\beta(N_v(t)/N_h(t))$, the probability that the mosquito is infected $I_v(t)/N_v(t)$, and the probability of successful transmission of infection $\theta_{vh}$, i.e.

$$\lambda_{vh}(t) = \beta\theta_{vh}\frac{I_v(t)}{N_h(t)}.$$

Without treatment, the force of infection from humans to mosquitoes, defined as

$$\lambda_{hv}^{wo}(t) = \beta\theta_{hv}\frac{I_h(t)}{N_h(t)}, \tag{2.1}$$

is just the product of the average number of bites per mosquito per unit time $\beta$, the probability that the human is infectious $I_h(t)/N_h(t)$, and the probability of successful transmission of infection $\theta_{hv}$. It is known that the antifilarial drugs given during MDA can instantaneously kill the microfilariae in humans and can halt the reproduction of adult worms, thus reducing the probability of transmission for a significant amount of time [63]. So, if we define $p$ as the proportion of reduction in transmission due to treatment, then the transmission probability becomes $\theta_{hv}(1-p)$. Hence, with treatment, the force of infection is given by

$$\lambda_{hv}^{w}(t) = \beta\theta_{hv}(1-p)\frac{I_h(t)}{N_h(t)}. \tag{2.2}$$

Now, suppose only a proportion $\gamma$ of the human population is given treatment. Then, the total force of infection from humans to mosquitoes is computed using equations (2.1) and (2.2) as follows:

$$\lambda_{hv}(t) = \lambda_{hv}^{wo}(t)(1-\gamma) + \lambda_{hv}^{w}(t)\gamma = \beta\theta_{hv}\frac{I_h(t)}{N_h(t)}(1-p\gamma).$$

Thus, upon sufficient bites to infectious humans, uninfected mosquitoes move to the infected class at the rate $\lambda_{hv}(t)$. Similarly, after adequate infective mosquito bites, uninfected humans move to the latent class at the rate $\lambda_{vh}(t)$. Note that the force of infection from mosquitoes to humans remains the same with or without treatment since the antifilarial drugs ingested by humans do not directly affect the mosquitoes. Owing to the growth and development of parasites in the human body, individuals in the latent class will eventually become infectious at the rate $\alpha$. Since there is no permanent nor temporary immunity, the latent and infectious humans given treatment move back to the uninfected class at the rates $r_1$ and $r_2$, respectively. We note that $r_1 \geq r_2$, since the level of infection (i.e. density and developmental stage of parasites in the body) of the latent individuals is assumed to be lower than those who are infectious.

We assume that the parameters $b_h$, $b_v$, $\delta_h$, $\delta_v$, $\theta_{vh}$, $\theta_{hv}$, $\beta$, $\alpha$ are strictly positive real numbers. We also assume that the antifilarial drugs are effective; thus, $r_1$, $r_2$ are strictly positive, and $p$ is within $(0, 1]$. Since the treatment coverage $\gamma$ is a controllable parameter, we assume that it can take any value within $[0, 1]$, where $\gamma = 0$ implies that no one in the population is given treatment while $\gamma = 1$ implies that the whole population is given treatment. We also assume that all individuals given treatment actually ingest the antifilarial drugs.

Figure 1 provides a graphical interpretation of our model. Based on our model descriptions and assumptions, the proposed model is governed by the following system of differential equations:

$$\left.\begin{aligned}
\frac{dU_h}{dt} &= b_h N_h + (r_1 L_h + r_2 I_h)\gamma - (\lambda_{vh} + \delta_h)U_h, \\
\frac{dL_h}{dt} &= \lambda_{vh}U_h - (\alpha + r_1\gamma + \delta_h)L_h, \\
\frac{dI_h}{dt} &= \alpha L_h - (r_2\gamma + \delta_h)I_h, \\
\frac{dU_v}{dt} &= b_v N_v - (\lambda_{hv} + \delta_v)U_v \\
\frac{dI_v}{dt} &= \lambda_{hv}U_v - \delta_v I_v,
\end{aligned}\right\} \tag{2.3}$$

and

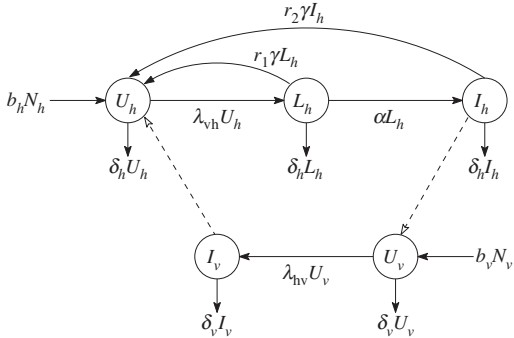

**Figure 1.** Diagram of the LF transmission model. Solid lines represent the transition of humans and mosquitoes between different states of infection. Dashed lines represent the transfer of parasites from human to mosquito and vice versa through a mosquito bite.

where

$$\lambda_{\mathrm{vh}} = \beta\theta_{\mathrm{vh}}\frac{I_v}{N_h}, \quad \lambda_{\mathrm{hv}} = \beta\theta_{\mathrm{hv}}(1 - p\gamma)\frac{I_h}{N_h}.$$

## 2.2. Epidemiological data

The available filariasis data from the Philippine Department of Health's Field Health Service Information System (FHSIS) Annual Reports [64] were compiled for each Philippine province and region covering the years 2009–2018. The 2009–2018 data report annual prevalence rates, computed as the percentage of microfilariae-positive individuals over the total cases examined in the area. According to the DOH FHSIS reports, 5 (out of 5) provinces and 3 (out of 6) cities in the administrative region of Caraga have been endemic for LF [64]. As of 2018, one province (Surigao del Norte) and one city (Surigao City) in Caraga Region are still implementing MDA for LF while the other four provinces have already been declared LF-free (Agusan del Sur and Dinagat Islands in 2010, Surigao del Sur in 2013, and Agusan del Norte in 2015). The 2018 DOH FHSIS report also suggests that there are still confirmed cases of LF in the three provinces and 1 city previously declared LF-free [64]. Hence, the Caraga filariasis data are used in our simulations. From the recorded total human population and annual prevalence rates, the infectious human population per year was estimated as the proportion of the total population that are microfilariae positive (i.e. $I_h$ = total population × prevalence rate). Table 1 gives a summary of our dataset.

## 2.3. Parameter values and initial conditions

The human natural death rate $\delta_h$, measured in weeks, is computed as the inverse of life expectancy, which is calculated by subtracting the median age of the Caraga population in 2010 [65] from the recorded life expectancy at birth in Caraga in 2010 [66]. An approximate for the obtained value is $\delta_h = 0.00042$ (per week). Using the obtained value and the Caraga population data for the years 2009–2018 in table 1, we estimate the human birth rate $b_h$ from the exact solution of $\mathrm{d}N_h/\mathrm{d}t = (b_h - \delta_h)N_h$ which is $N_h(t) = N_h(0)\,\mathrm{e}^{(b_h - \delta_h)t}$. Solving this numerically using the built-in Matlab function `lsqcurvefit`, we find that the birth rate is approximately $b_h = 0.0006$ (per week).

Meanwhile, since LF and dengue are transmitted by mosquitoes of the same genus (*Aedes*), the parameter values for the mosquito death rate $\delta_v$ and mosquito biting rate $\beta$ are obtained from a dengue model presented by de los Reyes V & Escaner IV [67]. Here, we assume that $b_v = \delta_v$.

The progression from $L_h$ to $I_h$ ($1/\alpha$) is usually between 6 and 12 months [62], but we fix $\alpha = 0.0288$ which is about eight months, following a study by Jambulingam *et al.* [44]. In [39], Norman *et al.* set a value of $1.13 \times 10^{-4}$ for the proportion of L3 filarial parasites entering a host which develop into adult worms. In our model, this value is assigned to the transmission probability from vector to human, $\theta_{\mathrm{vh}}$. In the same paper, the proportion of mosquitoes which pick up infection when biting an infected host was assigned a value of 0.37, which we set to be the transmission probability from human to vector, $\theta_{\mathrm{hv}}$.

The treatment coverage $\gamma = 0.619$ is computed as the average of the recorded MDA programme coverages in Caraga from 2010 to 2018 obtained from the DOH FHSIS data [64]. The treatment rates $r_1$ and $r_2$ in table 1 are defined as the inverse of the average duration of the treatment of humans in compartments $L_h$ and $I_h$,

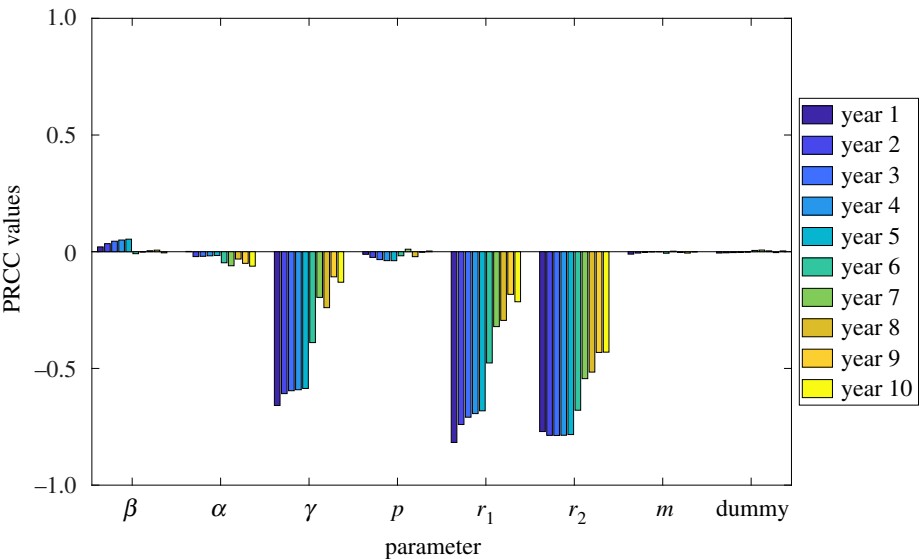

**Figure 2.** PRCC values depicting the sensitivities of the model output $L_h + I_h$ with respect to the model parameters at 10 different time points.

**Table 1.** The DOH FHSIS filariasis data in Caraga Region from 2009 to 2018. The data under total population, total cases examined, cases found positive and prevalence rate are obtained from the DOH FHSIS Annual Reports [64] while the data under infectious population are computed by multiplying total population by prevalence rate.

| year | total population | total cases examined | cases found positive | prevalence rate (%) | infectious population |
|---|---|---|---|---|---|
| 2009 | 2 501 400 | 5 036 | 54 | 1.07 | 26 822 |
| 2010 | 2 429 224 | 8 275 | 174 | 2.10 | 51 080 |
| 2011 | 2 611 700 | 3 984 | 25 | 0.63 | 16 389 |
| 2012 | 2 507 410 | 18 406 | 189 | 1.03 | 25 747 |
| 2013 | 2 544 172 | 18 406 | 189 | 1.03 | 26 125 |
| 2014 | 2 581 399 | 11 537 | 40 | 0.35 | 8950 |
| 2015 | 2 619 098 | 39 831 | 25 | 0.06 | 1644 |
| 2016 | 2 657 380 | 35 033 | 145 | 0.41 | 10 999 |
| 2017 | 2 828 583 | 49 744 | 73 | 0.15 | 4151 |
| 2018 | 2 694 944 | 42 943 | 56 | 0.13 | 3514 |

respectively, i.e. the time it takes for an infected person to be treated and to move back to the uninfected class. These parameters are estimated using the available filariasis data in Caraga.

The parameter $p$ represents the proportion of reduction in transmission due to treatment. From what has been inferred from the literature, this parameter depends on many factors such as the percentage of microfilariae and adult worms killed due to treatment, the reduction in Mf production, and the duration of the sustained reduction in the density of these parasites in the human body following treatment. Here, a value of 0.6 is assigned to the parameter $p$. We show in the next section that the infected human population $L_h + I_h$ is not sensitive to the parameter $p$ (figure 2).

In our numerical simulations, we use the 2009 population data from Caraga as initial conditions. Since only the total human population $N_h(0)$ and the infectious population $I_h(0)$ are available, it is assumed that the remaining human population, $N_h(0) - I_h(0)$, can be apportioned between the uninfected $U_h(0)$ and latent $L_h(0)$ populations using a partitioning parameter $m \in [0, 1]$ in the relation

and
$$\left. \begin{array}{l} U_h(0) = (N_h(0) - I_h(0)) \times m \\ L_h(0) = (N_h(0) - I_h(0)) \times (1 - m), \end{array} \right\} \qquad (2.4)$$

**Table 2.** Parameters of the LF transmission model.

| parameter | description | value per week | references |
|---|---|---|---|
| $b_h$ | human birth rate | 0.0006 | data-fitted[a] |
| $\delta_h$ | human natural death rate | 0.00042 | calculated [65,66] |
| $b_v$ | mosquito birth rate | 0.1 | assumed |
| $\delta_v$ | mosquito natural death rate | 0.1 | [67] |
| $\beta$ | mosquito biting rate | 1 | [67] |
| $\theta_{vh}$ | probability of transmission from mosquito to human | 0.000113 | [39] |
| $\theta_{hv}$ | probability of transmission from human to mosquito | 0.37 | [39] |
| $\alpha$ | progression rate from $L_h$ to $I_h$ | 0.0288 | [44] |
| $r_1$ | treatment rate of $L_h$ | 0.430848 | data-fitted[b] |
| $r_2$ | treatment rate of $I_h$ | 0.010038 | data-fitted[b] |
| $\gamma$ | treatment coverage | 0.619 | [64] |
| $p$ | proportion of reduction in transmission due to treatment | 0.6 | assumed |
| $m$ | partitioning parameter for the initial population | 0.853284 | data-fitted[b] |

[a]From the Caraga population data.
[b]From the Caraga filariasis data.

where the obtained values are rounded off to the nearest whole number to be biologically consistent with the human population. This parameter $m$ is also estimated along with $r_1$ and $r_2$ using the available filariasis data from Caraga. Meanwhile, the initial values for the mosquito population are assumed to be $U_v(0) = 1\,000\,000$ and $I_v(0) = 100\,000$. Table 2 lists the parameter values of the LF transmission model.

# 3. Results and discussion

## 3.1. Mathematical analysis of the model

Since all model parameters and state variables are assumed to be non-negative, the total populations are also non-negative, i.e. $N_h(t) = U_h(t) + L_h(t) + I_h(t) \geq 0$ and $N_v(t) = U_v(t) + I_v(t) \geq 0$. Hence, our region of biological interest is

$$\Psi = \left\{ (U_h, L_h, I_h, U_v, I_v) \in \mathbb{R}^5_{\geq 0} \right\}.$$

The model system in equation (2.3) is mathematically and epidemiologically well posed in $\Psi$.

**Theorem 3.1.** *System (2.3) has two steady-state solutions:*

(i) *the disease-free equilibrium $E_0 = (b_h N_h / \delta_h, 0, 0, b_v N_v / \delta_v, 0)$ and*
(ii) *the endemic equilibrium $E^* = (U_h^*, L_h^*, I_h^*, U_v^*, I_v^*)$, where*

$$U_h^* = \frac{b_h N_h (\alpha + r_1 \gamma + \delta_h)(r_2 \gamma + \delta_h)}{\delta_h [(\alpha + r_1 \gamma + \delta_h)(r_2 \gamma + \delta_h) + \lambda_{vh}^* (\alpha + r_2 \gamma + \delta_h)]},$$

$$L_h^* = \frac{\lambda_{vh}^* b_h N_h (r_2 \gamma + \delta_h)}{\delta_h [(\alpha + r_1 \gamma + \delta_h)(r_2 \gamma + \delta_h) + \lambda_{vh}^* (\alpha + r_2 \gamma + \delta_h)]},$$

$$I_h^* = \frac{\lambda_{vh}^* \alpha b_h N_h}{\delta_h [(\alpha + r_1 \gamma + \delta_h)(r_2 \gamma + \delta_h) + \lambda_{vh}^* (\alpha + r_2 \gamma + \delta_h)]},$$

$$U_v^* = \frac{b_v N_v [\lambda_{vh}^* \delta_h (\alpha + r_2 \gamma + \delta_h) + \delta_h (\alpha + r_1 \gamma + \delta_h)(r_2 \gamma + \delta_h)]}{\lambda_{vh}^* [\alpha \beta \theta_{hv} (1 - p\gamma) b_h + \delta_h \delta_v (\alpha + r_2 \gamma + \delta_h)] + \delta_v \delta_h (\alpha + r_1 \gamma + \delta_h)(r_2 \gamma + \delta_h)}$$

*and*  $$I_v^* = \frac{\lambda_{vh}^* \alpha \beta \theta_{hv} (1 - p\gamma) b_h b_v N_v}{\lambda_{vh}^* \delta_v [\alpha \beta \theta_{hv} (1 - p\gamma) b_h + \delta_h \delta_v (\alpha + r_2 \gamma + \delta_h)] + \delta_v^2 \delta_h (\alpha + r_1 \gamma + \delta_h)(r_2 \gamma + \delta_h)},$$

*where*

$$\lambda_{\mathrm{vh}}^* = \frac{\alpha\beta^2\theta_{\mathrm{vh}}\theta_{\mathrm{hv}}(1-p\gamma)b_h b_v N_v - \delta_v^2\delta_h N_h(\alpha+r_1\gamma+\delta_h)(r_2\gamma+\delta_h)}{\delta_v N_h[\alpha\beta\theta_{\mathrm{hv}}(1-p\gamma)b_h + \delta_v\delta_h(\alpha+r_2\gamma+\delta_h)]}.$$

Since $E_0$ is a steady-state solution wherein the infected populations are zero, $E_0$ is referred to as the *disease-free equilibrium* (DFE) solution. The steady-state solution $E^*$ is known as the *endemic equilibrium* solution since the infection is constantly maintained in the population. The derivations of $E_0$ and $E^*$ are discussed in detail in appendix A.

The basic reproduction number, $\mathcal{R}_0$, is a threshold parameter that is used to assess whether or not a disease will invade a population. In theory, if $\mathcal{R}_0 < 1$, then each infected person produces less than one new infected individual in their entire period of infectiousness, which implies that the infection will not be sustained in the population. On the contrary, if $\mathcal{R}_0 > 1$, then each infected individual infects more than one person implying that the disease will eventually invade the population. Here, the basic reproduction number is computed using the method of next-generation matrix, a method introduced by Diekmann *et al.* [68]. The value of $\mathcal{R}_0$ for the model system (2.3) is presented in the next theorem. For the proof, we refer readers to appendices B and C.

**Theorem 3.2.** *The basic reproduction number for the model system (2.3) is*

$$\mathcal{R}_0 = \sqrt{\frac{\alpha\beta^2\theta_{\mathrm{vh}}\theta_{\mathrm{hv}}(1-p\gamma)b_h b_v N_v}{\delta_v^2\delta_h N_h(\alpha+r_1\gamma+\delta_h)(r_2\gamma+\delta_h)}}.$$

*Moreover, the DFE $E_0$ is locally asymptotically stable when $\mathcal{R}_0 < 1$ and unstable when $\mathcal{R}_0 > 1$. The endemic equilibrium $E^*$ is locally asymptotically stable when $\mathcal{R}_0 > 1$.*

## 3.2. Sensitivity analysis and parameter estimation

Sensitivity analysis is a tool used to identify and rank critical input parameters in a model with respect to their impact on the reference model output. An input parameter is said to be influential if small variations of its value result to significant changes in the model output. Based on the result of sensitivity analysis, one can have an idea which input parameters need to be assigned accurate values (i.e. the most influential parameters) and which ones can be roughly estimated (i.e. the less influential parameters). In this study, we used one of the most efficient and reliable global sensitivity analysis techniques—the partial rank correlation coefficient (PRCC) method combined with the Latin hypercube sampling (LHS) technique for the sampling of parameters [69,70]. Numerical simulations for sensitivity analysis using the LHS/PRCC method were carried out using modified versions of the PRCC Matlab codes presented by Marino *et al.* [69].

The sign and magnitude of the PRCC values, which lie between −1 and +1, characterize the qualitative relationship between the model input and the model output. A positive PRCC implies a positive correlation between the input and the output; that is, an increase in the model input will result in an increase in the model output. On the other hand, a negative PRCC implies a negative correlation wherein an increase in the model input causes a decrease in the model output, and vice versa. The absolute value of a PRCC measures the importance of the model input to the relative model output. The greater the magnitude of a PRCC value, the greater the impact of the input to the output. The *p*-value of the PRCC indicates the statistical significance of the value. In most cases, a parameter is said to be sensitive or influential to the model output if the magnitude of the PRCC is greater than or equal to 0.5 and the corresponding *p*-value is less than 0.05 [69]. We used these criteria to determine the influential parameters in our model.

We tested the sensitivity of the model parameters $\beta$, $\alpha$, $\gamma$, $p$, $r_1$, $r_2$, $m$ and a dummy parameter, with respect to the infected human population $L_h + I_h$. A dummy parameter is included to test the robustness of the model as in [69]. That is, since it does not appear in the model equations and does not affect the model in any other way, the dummy parameter should be assigned a sensitivity index of zero in the simulations. The sample size is set to $N = 10\,000$ and the time points of interest are every year for a duration of 10 years or every after 52 weeks for $T = 520$ weeks.

We fix the values of $b_h$ and $\delta_h$ based on the calculations using the available data. We also fix the values of $b_v$, $\delta_v$, and the transmission probabilities $\theta_{\mathrm{vh}}$ and $\theta_{\mathrm{hv}}$, following the results in [39,67].

**Table 3.** Results of parameter estimation using `lsqcurvefit` with the bounds used. The estimate is computed as the average of the estimates from 1000 distinct initial guesses generated by the LHS method.

| parameter | estimate | lower bound | upper bound |
|---|---|---|---|
| $r_1$ | 0.430848 | 0.0019 | 0.5 |
| $r_2$ | 0.010038 | 0.0019 | 0.5 |
| $m$ | 0.853284 | 0 | 1 |

Since there is limited information on the values for $\alpha$, $\gamma$ and $p$, a uniform distribution is assigned to their corresponding parameter ranges which we set to $\pm 50\%$ of their respective baseline values given in table 2. On the other hand, since $\beta$ has already been estimated in [67], we used a normal distribution with the nominal value in table 2 as the mean and a standard deviation of 0.1. The range of values for the treatment rates $r_1$ and $r_2$ is from 0.0019 to 0.5, or from two weeks to 10 years, while the range of values for the partitioning parameter $m$ is from 0 to 1. Both ranges are assigned a uniform distribution.

The result of our simulations is shown in figure 2, where each coloured bar graph corresponds to a particular time point. Based on our criteria for determining critical parameters, the results suggest that the parameters $\gamma$, $r_1$ and $r_2$ are the most influential parameters throughout the running time of 10 years or 520 weeks. It can be observed that $\gamma$, $r_1$ and $r_2$ have a negative correlation to the model output $L_h + I_h$. This implies that an increase in the treatment coverage and faster treatment rates will most likely result in a decrease in the total infected human population. We also observed that all three parameters have high magnitude of PRCC values for the first five years, after which the magnitude of the PRCC values of $\gamma$ and $r_1$ decreased below 0.5. The PRCC values of $r_2$ decreased below 0.5 in the last two years.

In our parameter estimation, we opted not to include the treatment coverage $\gamma$ since there are already recorded values for this in the DOH FHSIS filariasis data [64]. This left us with the parameters $r_1$ and $r_2$, which we estimate together with the partitioning parameter $m$, using the available population data from Caraga Region [64] covering the years 2009–2018.

From the DOH FHSIS filariasis data in Caraga Region in table 1, we have 10 data points for $I_h$ (2009–2018) with the 2009 data as the initial condition. The parameters $r_1$, $r_2$ and $m$ are estimated by minimizing the mean of the squared difference between the available data and the model output at the corresponding time point using the Matlab function `lsqcurvefit`.

To establish unbiased choice of initial conditions for the parameter estimation, the LHS method is used to generate 1000 sets of initial guesses for the estimates. For each set of initial guesses, parameters are estimated using `lsqcurvefit`. The 'best' estimate is then computed as the average of the 1000 estimates from 1000 distinct initial guesses. The simulation results, along with the lower bound and upper bound set for each of the three parameters for the Matlab simulations are given in table 3.

Using the estimated value for the partitioning parameter $m = 0.853284$ and the 2009 filariasis data from Caraga in table 1, we can now solve for the initial populations for $U_h$ and $L_h$ in equation (2.4). Meanwhile, the estimated value for the treatment rate $r_1$ is 0.430848 which means that the treatment period $1/r_1$ is about 16 days, and the estimated value for the treatment rate $r_2$ is 0.010038 which implies that $1/r_2$ is about 2 years. Recall that the progression from the latent stage to the infectious stage lasts for about 6–12 months [62]. Hence, the obtained value for $r_1$ highlights the effectiveness of MDA in preventing new infections. The obtained value for $r_2$ explains that although transmission of L3 larvae from mosquitoes to humans is inefficient, the treatment of infectious individuals takes a long time because of the 5-year reproductive life span of adult filarial worms [61]. This highlights the importance of maintaining a high treatment coverage since majority of the infected population are asymptomatic. Figure 3 depicts the available filariasis data plotted with the corresponding model fit from parameter estimation.

Parameter bootstrapping is also used to investigate the reliability of our estimated values. Bootstrapping is a statistical technique used to quantify uncertainty and construct confidence intervals in parameter estimates [71]. Following the algorithm presented by Chowell [71], 1000 samples of synthetic datasets are generated from the best-fit model by assuming a Poisson error structure. Parameters are then re-estimated from each synthetic dataset using `lsqcurvefit` to obtain a new set

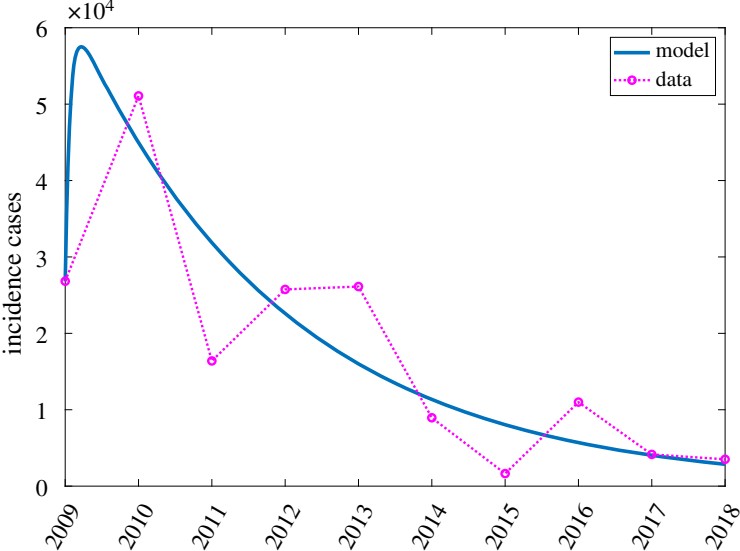

**Figure 3.** The available data points for $I_h$ and the corresponding fit from parameter estimation.

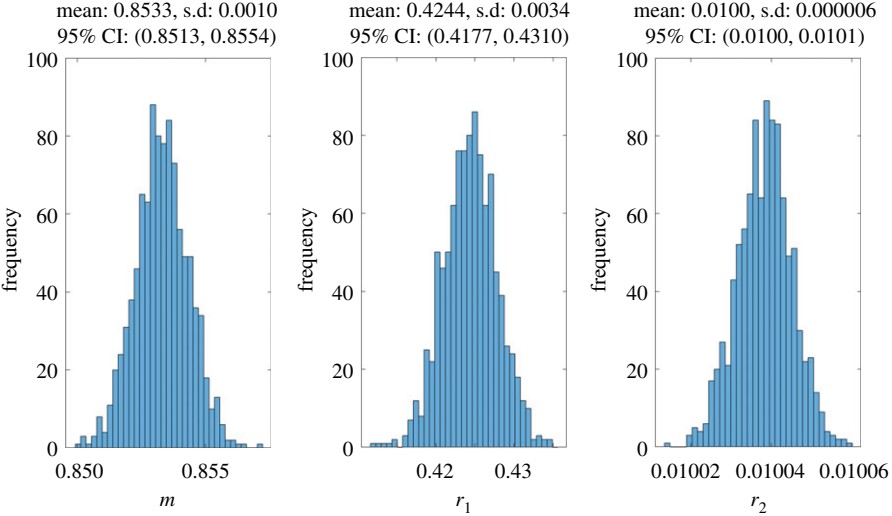

**Figure 4.** Results of parameter bootstrapping with the mean, standard deviation (s.d.) and 95% confidence interval (CI).

of parameter estimates. Numerical simulations for parameter bootstrapping were carried out using modified versions of the sample Matlab codes presented by Chowell [71]. Figure 4 shows the parameter bootstrapping results with the means, standard deviations and 95% confidence intervals for each parameter. We observe low standard deviations and narrow ranges of the 95% confidence intervals for all the parameters. Moreover, the estimates obtained for $m$, $r_1$ and $r_2$ (table 3) all lie within their corresponding confidence intervals.

### 3.3. Optimal control strategy

In implementing MDA programmes, the treatment coverage can vary through time depending on the efforts of the government. In the LF transmission model, this translates to the constant parameter $\gamma$ becoming a function of time $u(t)$. Since the main purpose of MDA is to limit and reduce the infected humans by interrupting the transmission of infection, the proposed optimal control problem is to minimize the number of infected (both latent and infectious) hosts as well as

the corresponding cost of implementing MDA. Here, minimizing the infected mosquito population is not included in the objective since there is no control directly affecting the vector population. From our simulations, we aim to gain insights about the optimal MDA coverage over a duration of $T$ years, $T = (t_f - t_0) > 0$, such that the objective is satisfied. Hence, our objective functional to be minimized is

$$J(u) = \int_{t_0}^{t_f} \left[ L_h(t) + I_h(t) + \frac{c}{2} u^2(t) \right] dt,$$

where $t_0$ and $t_f$ are taken as 2018 and 2030, respectively. Here, $c$ is a weighting parameter associated with the implementation cost of MDA. We note that the parameter $c$ does not represent *per se* the actual monetary cost of implementing MDA. Instead, it is a constant parameter that balances the size and importance of each term in the integrand. That is, if $c$ is too high, more importance will be given in reducing the cost of implementation of MDA compared to the infected population. On the other hand, if $c$ is low, the minimization problem will put equal importance in minimizing the infectious population and the cost of implementation. It is assumed that the control is a quadratic function to represent nonlinear implementation costs. Our goal is to find optimal $u^*$ satisfying

$$J(u^*) = \min_u J(u)$$

subject to

$$\frac{dU_h}{dt} = b_h N_h + (r_1 L_h + r_2 I_h)u(t) - \beta\theta_{\mathrm{vh}}\frac{I_v}{N_h}U_h - \delta_h U_h,$$

$$\frac{dL_h}{dt} = \beta\theta_{\mathrm{vh}}\frac{I_v}{N_h}U_h - (\alpha + r_1 u(t) + \delta_h)L_h,$$

$$\frac{dI_h}{dt} = \alpha L_h - (r_2 u(t) + \delta_h)I_h,$$

$$\frac{dU_v}{dt} = b_v N_v - \beta\theta_{\mathrm{hv}}(1 - pu(t))\frac{I_h}{N_h}U_v - \delta_v U_v$$

and

$$\frac{dI_v}{dt} = \beta\theta_{\mathrm{hv}}(1 - pu(t))\frac{I_h}{N_h}U_v - \delta_v I_v,$$

with the initial conditions $U_h(0) = U_{h,0}$, $L_h(0) = L_{h,0}$, $I_h(0) = I_{h,0}$, $U_v(0) = U_{v,0}$, $I_v(0) = I_{v,0}$, and such that $u_{\min} \le u(t) \le u_{\max}$.

Pontryagin's maximum principle [72] transforms the optimal control problem into a problem that minimizes a Hamiltonian $H$ pointwise with respect to the control $u(t)$. Using $x(t) = [U_h, L_h, I_h, U_v, I_v](t)$ and $\lambda(t) = [\lambda_1, \ldots, \lambda_5](t)$, the Hamiltonian $H$ is formed as

$$H(t, x(t), u(t), \lambda(t)) = L_h + I_h + \frac{c}{2}u^2$$

$$+ \lambda_1 \left[ b_h N_h + (r_1 L_h + r_2 I_h)u(t) - \beta\theta_{\mathrm{vh}}\frac{I_v}{N_h}U_h - \delta_h U_h \right]$$

$$+ \lambda_2 \left[ \beta\theta_{\mathrm{vh}}\frac{I_v}{N_h}U_h - (\alpha + r_1 u(t) + \delta_h)L_h \right]$$

$$+ \lambda_3 [\alpha L_h - (r_2 u(t) + \delta_h)I_h]$$

$$+ \lambda_4 \left[ b_v N_v - \beta\theta_{\mathrm{hv}}(1 - pu(t))\frac{I_h}{N_h}U_v - \delta_v U_v \right]$$

$$+ \lambda_5 \left[ \beta\theta_{\mathrm{hv}}(1 - pu(t))\frac{I_h}{N_h}U_v - \delta_v I_v \right].$$

Applying Pontryagin's maximum principle, we obtain the following result. The proof of this theorem can be found in appendix D.

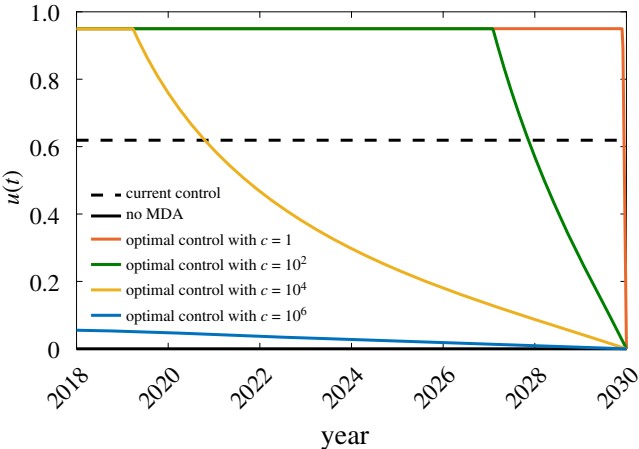

**Figure 5.** Control profile.

**Theorem 3.3.** *There exists optimal control $u^*(t)$ minimizing the objective functional $J(u(t))$ over $\Omega = \{u_{\min} \leq u \leq u_{\max}, u \in \mathcal{L}^2(2018, 2030)\}$. Given this optimal solution, there exist adjoint variables $\lambda_1(t), \ldots, \lambda_5(t)$, which satisfy*

$$\frac{d\lambda_1}{dt} = (\lambda_1 - \lambda_2)\beta\theta_{vh}\frac{I_v}{N_h} + \lambda_1\delta_h,$$

$$\frac{d\lambda_2}{dt} = -1 + (\lambda_2 - \lambda_1)r_{1u} + (\lambda_2 - \lambda_3)\alpha + \lambda_2\delta_h,$$

$$\frac{d\lambda_3}{dt} = -1 + (\lambda_3 - \lambda_1)r_{2u} + (\lambda_4 - \lambda_5)\beta\theta_{hv}(1 - pu)\frac{U_v}{N_h} + \lambda_3\delta_h,$$

$$\frac{d\lambda_4}{dt} = (\lambda_4 - \lambda_5)\beta\theta_{hv}(1 - pu)\frac{I_h}{N_h} + \lambda_4\delta_v$$

*and*

$$\frac{d\lambda_5}{dt} = (\lambda_1 - \lambda_2)\beta\theta_{vh}\frac{U_h}{N_h} + \lambda_5\delta_v,$$

*with transversality conditions $\lambda_i(t_f) = 0$, for $i = 1, \ldots, 5$. Furthermore, the optimality equation is as follows:*

$$u^*(t) = \min\left(u_{\max}, \left(u_{\min}, \frac{1}{c}\left[r_1 L_h(\lambda_2 - \lambda_1) + r_2 I_h(\lambda_3 - \lambda_1) + \beta\theta_{hv}p\frac{I_h U_v}{N_h}(\lambda_5 - \lambda_4)\right]\right)\right).$$

Since our compiled population data in table 1 suggest that there are still infectious individuals in Caraga in 2018, we look at the disease dynamics after 2018 to obtain insights on the optimal strategy for implementing MDA to eliminate LF in the population by 2030. Hence, the choice of $t_0 = 2018$ and $t_f = 2030$ in the objective functional.

Numerical simulations were carried out in Matlab using the forward–backward sweep method [60]. We set $u_{\min} = 0$ to account for the possibility that MDA is stopped after 2018, and $u_{\max} = 0.95$ as we recognize that a 100% treatment coverage is quite difficult to achieve in reality. We also considered different values for the weighting parameter $c$ to investigate how the variations in the MDA implementation cost affect the optimal control solution. The obtained optimal control solutions for $c = 10^i$, $i = 0, 2, 4, 6$, were compared with the no control solution ($u(t) = 0$, $\forall t$) and the constant control solution ($u(t) = 0.619$, $\forall t$). Figures 5 and 6 show the obtained control profile and the corresponding effect in the dynamics of the infectious population $I_h(t)$.

Observe from figure 6 that if MDA is stopped after 2018, the decrease in the infectious population over a span of 12 years is only about 15%. On the other hand, if the current control is maintained for another 12 years the infectious population can decrease further by 98%.

One apparent conclusion from the optimal control solutions in figure 5 is that the MDA coverage is inversely proportional to the implementation cost. That is, the lower the cost, the higher the MDA coverage. Consequently, a higher MDA coverage leads to a lower number of infected individuals at the end time as illustrated in figure 6. Here, we compare the optimal control results for different values of $c$ to the no MDA strategy. As recorded in table 4, with $c$ values equal to $10^0$, $10^2$ or $10^4$, the $I_h$ population can be reduced by at least 91% at the end of the control period. When $c = 10^6$, we observe that the optimal control problem gives more emphasis on minimizing the MDA implementation cost compared

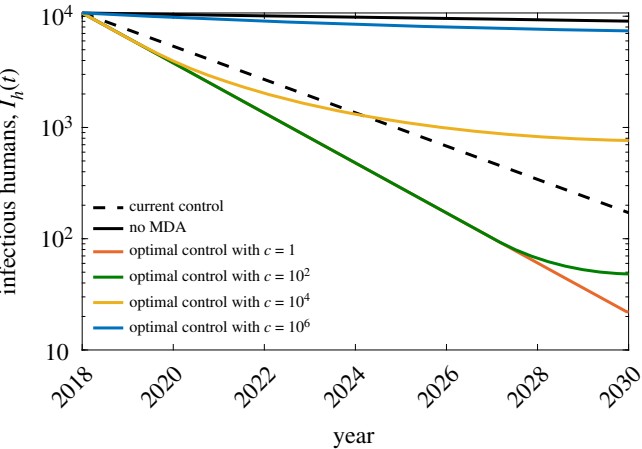

**Figure 6.** The corresponding effect of the controls in the dynamics of the infectious population $I_h$. The y-axis is in log scale to better illustrate the difference in the dynamics towards the end time for different values of $c$.

**Table 4.** Per cent reduction in the infectious population $I_h$ when the optimal control using implementation cost $c$ is applied. Per cent reduction is the relative difference in percentage between the number of infectious population in 2030 when optimal control is applied and without MDA.

| implementation cost $c$ | per cent reduction in $I_h$ (%) |
| --- | --- |
| $10^0$ | 99.76 |
| $10^2$ | 99.47 |
| $10^4$ | 91.55 |
| $10^6$ | 18.24 |

to minimizing the infected population. As a result, we obtain a low MDA coverage which led to a higher number of infected individuals at the end time. In this case, the optimal solution corresponds to only 18% reduction in the infectious population compared to the no MDA strategy. The optimal control solutions suggest that the infected population can be further minimized if MDA coverage is scaled up. Even though we do not have enough information on the actual cost of MDA implementation, the data on MDA coverage in the Philippines since 2001 suggest that scaling up MDA programmes is possible especially since only 6 out of 46 endemic provinces are implementing MDA as of 2018.

# 4. Conclusion

LF remains a public health problem in the Philippines with millions of Filipinos still at risk of being infected as of 2018. In this work, we developed a mathematical model of LF transmission involving the interaction of human and mosquito populations. Using this model, we investigated how the implementation of the annual MDA for LF affects the disease dynamics in the LF-endemic region of Caraga. Sensitivity analysis using the LHS/PRCC method showed that the infected human population is most sensitive to the treatment coverage (i.e. how much of the population receives treatment) and the treatment rates (i.e. how effective the antifilarial drugs are in reducing the parasite density in infected humans). This highlights the importance of strategic MDA implementation, and efficient data gathering and data reporting to obtain more accurate simulation results and produce more realistic and relevant insights. The treatment parameters were estimated using the available filariasis data in Caraga obtained from the Philippine DOH's FHSIS. The estimated values emphasize the importance of MDA in preventing new infections. Parameter bootstrapping showed small variability in all the parameter estimates, indicated by the low standard deviations and narrow confidence intervals.

The application of optimal control theory highlighted the importance of maintaining a high treatment coverage per year to effectively minimize the infected population by the year 2030. To achieve this, it is

important that there is an effective and systematic implementation of MDA from the national level down to the implementation units. Activities such as the distribution of antifilarial drugs, collection of data, and diagnostic examinations must be enhanced. To address the problems with MDA acceptance [73], educational activities must be conducted before the distribution of antifilarial drugs to inform the people and help them understand how participating in MDA will benefit them and their communities. Field workers and volunteers must also be well informed about the MDA programme and the disease. Most importantly, since LF is closely related to poverty, eliminating the disease requires the active commitment of the government to improve the lives of its poorest and most vulnerable citizens. This does not only include the distribution of treatment against LF and other NTDs but also the provision of better living conditions by giving people better opportunities for employment, proper education, clean water and other basic human needs.

One limitation of our study is the uncertainty in the values of our model inputs, both initial conditions and model parameters. Since the model is applied specifically to Caraga, the parameter values must be accurate to well represent the transmission dynamics of LF in the region. However, majority of the existing studies on LF in the Philippines are outdated and may no longer give accurate information on the disease dynamics at present. We urge the Philippine DOH to have a systematic, correct, and timely data gathering that may be relevant to the study of not only LF but also other diseases. We also encourage local researchers to look into LF in the country. One possible topic to explore is the dynamics of *Aedes poicilius* mosquitoes, or other known mosquito species that transmit LF in the Philippines, and review values of relevant parameters. Both clinical and field studies on the efficacy of the antifilarial drugs, or a comprehensive study on the general effects of MDA in reducing transmission in a population under varying circumstances, could also be helpful for this study. A possible extension of our model is the inclusion of the parasite dynamics, such as parasite aggregation, to better capture the transmission processes of LF. Future works could also focus on calibrating the model to incorporate heterogeneity in susceptibility and infectivity of individuals in the population. We also recommend cost analysis of MDA in the remaining endemic areas, such as the work of Amarillo *et al.* in Sorsogon, Philippines in 2009 [74].

Data accessibility. The filariasis data of the total population, total cases examined, cases found positive, and prevalence rate for Caraga Region, the Philippines, presented in table 1 are obtained from the Philippine Department of Health's Field Health Service Information System Annual Reports [64].
Authors' contributions. P.K.N.S. participated in the conceptualization, methodology, and formal analysis of the study, curated the data, carried out the simulations and computations, drafted the manuscript, acquired funding, and participated in the critical review of the manuscript; V.M.P.M. and R.G.M. participated in the conceptualization, methodology, and formal analysis of the study, drafted the manuscript, acquired funding, supervised the study, and participated in the critical review of the manuscript; V.Y.B.J. participated in the conceptualization and formal analysis of the study, supervised the study, and participated in the critical review of the manuscript. All authors participated in the final review and editing of the manuscript, and gave final approval for publication and agree to be held accountable for the work performed therein.
Competing interests. The authors declare no conflict of interest.
Funding. This work was funded by the Natural Sciences Research Institute, University of the Philippines Diliman under research grant nos. MAT-20-1-02 and MAT-21-1-04. The funders had no role in the design of the study; in the collection, analyses or interpretation of data; in the writing of the manuscript; or in the decision to publish the results.

# Appendix A. Computation of the steady-state solutions of the LF model

To find steady-state solutions of the system

$$\frac{dU_h}{dt} = b_h N_h + (r_1 L_h + r_2 I_h)\gamma - (\lambda_{vh} + \delta_h)U_h,$$

$$\frac{dL_h}{dt} = \lambda_{vh} U_h - (\alpha + r_1\gamma + \delta_h)L_h,$$

$$\frac{dI_h}{dt} = \alpha L_h - (r_2\gamma + \delta_h)I_h,$$

$$\frac{dU_v}{dt} = b_v N_v - (\lambda_{hv} + \delta_v)U_v$$

and

$$\frac{dI_v}{dt} = \lambda_{hv} U_v - \delta_v I_v,$$

where

$$\lambda_{\mathrm{vh}} = \beta\theta_{\mathrm{vh}}\frac{I_v}{N_h}, \quad \lambda_{\mathrm{hv}} = \beta\theta_{\mathrm{hv}}(1-p\gamma)\frac{I_h}{N_h},$$

we look for the corresponding solution to

$$\frac{dU_h}{dt} = \frac{dL_h}{dt} = \frac{dI_h}{dt} = \frac{dU_v}{dt} = \frac{dI_v}{dt} = 0.$$

We have the following:

$$\frac{dU_h}{dt} = 0 \Rightarrow U_h = \frac{b_h N_h + (r_1 L_h + r_2 I_h)\gamma}{\lambda_{\mathrm{vh}} + \delta_h}, \tag{A 1}$$

$$\frac{dL_h}{dt} = 0 \Rightarrow L_h = \frac{\lambda_{\mathrm{vh}} U_h}{\alpha + r_1\gamma + \delta_h}, \tag{A 2}$$

$$\frac{dI_h}{dt} = 0 \Rightarrow I_h = \frac{\alpha L_h}{r_2\gamma + \delta_h}, \tag{A 3}$$

$$\frac{dU_v}{dt} = 0 \Rightarrow U_v = \frac{b_v N_v}{\lambda_{\mathrm{hv}} + \delta_v} \tag{A 4}$$

and

$$\frac{dI_v}{dt} = 0 \Rightarrow I_v = \frac{\lambda_{\mathrm{hv}} U_v}{\delta_v}. \tag{A 5}$$

We first solve the state variables in terms of $\lambda_{\mathrm{vh}}$, then express $\lambda_{\mathrm{vh}}$ in terms of the model parameters. Using equation (A 4) and equation (A 5), we get

$$I_v = \frac{b_v N_v \lambda_{\mathrm{hv}}}{\delta_v(\lambda_{\mathrm{hv}} + \delta_v)}. \tag{A 6}$$

Moreover, if we plug-in equation (A 2) to equation (A 3), we get

$$I_h = \frac{\alpha\lambda_{\mathrm{vh}} U_h}{(r_2\gamma + \delta_h)(\alpha + r_1\gamma + \delta_h)}. \tag{A 7}$$

Plugging-in equation (A 7) and equation (A 2) to equation (A 1) and doing a few algebraic manipulations, we have

$$U_h = \frac{b_h N_h(r_2\gamma + \delta_h)(\alpha + r_1\gamma + \delta_h)}{\delta_h[(r_2\gamma + \delta_h)(\alpha + r_1\gamma + \delta_h) + \lambda_{\mathrm{vh}}(\alpha + r_2\gamma + \delta_h)]}. \tag{A 8}$$

Using equation (A 8) in equation (A 2), we get

$$L_h = \frac{\lambda_{\mathrm{vh}} b_h N_h(r_2\gamma + \delta_h)}{\delta_h[(r_2\gamma + \delta_h)(\alpha + r_1\gamma + \delta_h) + \lambda_{\mathrm{vh}}(\alpha + r_2\gamma + \delta_h)]}. \tag{A 9}$$

Similarly, using equations (A 9) and (A 2), we obtain

$$I_h = \frac{\alpha\lambda_{\mathrm{vh}} b_h N_h}{\delta_h[(r_2\gamma + \delta_h)(\alpha + r_1\gamma + \delta_h) + \lambda_{\mathrm{vh}}(\alpha + r_2\gamma + \delta_h)]}. \tag{A 10}$$

Recall that $\lambda_{\mathrm{vh}} = \beta\theta_{\mathrm{vh}}\frac{I_v}{N_h}$. Using equation (A 6), we have

$$\lambda_{\mathrm{vh}} = \frac{\beta\theta_{\mathrm{vh}} b_v N_v}{\delta_v N_h} \cdot \frac{\lambda_{\mathrm{hv}}}{\lambda_{\mathrm{hv}} + \delta_v}. \tag{A 11}$$

In the same manner, we find an expression for $\lambda_{\mathrm{hv}}$ using equation (A 10):

$$\lambda_{\mathrm{hv}} = \frac{\lambda_{\mathrm{vh}}\alpha\beta\theta_{\mathrm{hv}}(1-p\gamma)b_h}{\lambda_{\mathrm{vh}}\delta_h(\alpha + r_2\gamma + \delta_h) + \delta_h(r_2\gamma + \delta_h)(\alpha + r_1\gamma + \delta_h)}.$$

Then we have

$$\lambda_{\mathrm{hv}} + \delta_v = \frac{\lambda_{\mathrm{vh}}[\alpha\beta\theta_{\mathrm{hv}}(1-p\gamma)b_h + \delta_v\delta_h(\alpha + r_2\gamma + \delta_h)] + \delta_v\delta_h(r_2\gamma + \delta_h)(\alpha + r_1\gamma + \delta_h)}{\lambda_{\mathrm{vh}}\delta_h(\alpha + r_2\gamma + \delta_h) + \delta_h(r_2\gamma + \delta_h)(\alpha + r_1\gamma + \delta_h)} \tag{A 12}$$

and

$$\frac{\lambda_{\mathrm{hv}}}{\lambda_{\mathrm{hv}} + \delta_v} = \frac{\lambda_{\mathrm{vh}}\alpha\beta\theta_{\mathrm{hv}}(1-p\gamma)b_h}{\lambda_{\mathrm{vh}}[\alpha\beta\theta_{\mathrm{hv}}(1-p\gamma)b_h + \delta_v\delta_h(\alpha + r_2\gamma + \delta_h)] + \delta_v\delta_h(r_2\gamma + \delta_h)(\alpha + r_1\gamma + \delta_h)}. \tag{A 13}$$

Using equation (A 12) and equation (A 4), we get

$$U_v = \frac{b_v N_v[\lambda_{vh}\delta_h(\alpha + r_2\gamma + \delta_h) + \delta_h(r_2\gamma + \delta_h)(\alpha + r_1\gamma + \delta_h)]}{\lambda_{vh}[\alpha\beta\theta_{hv}(1 - p\gamma)b_h + \delta_v\delta_h(\alpha + r_2\gamma + \delta_h)] + \delta_v\delta_h(r_2\gamma + \delta_h)(\alpha + r_1\gamma + \delta_h)}. \tag{A 14}$$

Furthermore, plugging-in equation (A 13) to equation (A 6), we get

$$I_v = \frac{\lambda_{vh}\alpha\beta\theta_{hv}(1 - p\gamma)b_h b_v N_v}{\lambda_{vh}\delta_v[\alpha\beta\theta_{hv}(1 - p\gamma)b_h + \delta_v\delta_h(\alpha + r_2\gamma + \delta_h)] + \delta_v^2\delta_h(r_2\gamma + \delta_h)(\alpha + r_1\gamma + \delta_h)}. \tag{A 15}$$

Plugging-in equation (A 12) to equation (A 11) and after doing a few algebraic computations, we obtain

$$\lambda_{vh}(B\lambda_{vh} - A) = 0,$$

where

$$A = \alpha\beta^2\theta_{vh}\theta_{hv}(1 - p\gamma)b_h b_v N_v - \delta_v^2\delta_h N_h(\alpha + r_1\gamma + \delta_h)(r_2\gamma + \delta_h)$$

and

$$B = \delta_v N_h[\alpha\beta\theta_{hv}(1 - p\gamma)b_h + \delta_v\delta_h(\alpha + r_2\gamma + \delta_h)].$$

Here, we have two possible cases: either $\lambda_{vh} = 0$ or $B\lambda_{vh} - A = 0$.

$\quad$*Case 1.* Suppose $\lambda_{vh} = 0$. Then it follows from equations (A 9), (A 10), and (A 8), respectively, that $L_h = 0$, $I_h = 0$ and $U_h = b_h N_h/\delta_h$. It also follows from equations (A 15) and (A 14) that $I_v = 0$ and $U_v = \frac{b_v N_v}{\delta_v}$. Hence, we have the solution

$$E_0 = \left(\frac{b_h N_h}{\delta_h}, 0, 0, \frac{b_v N_v}{\delta_v}, 0\right).$$

$\quad$*Case 2.* Suppose $B\lambda_{vh} - A = 0$ and $\lambda_{vh} \neq 0$. Then

$$\lambda_{vh} = \frac{\alpha\beta^2\theta_{vh}\theta_{hv}(1 - p\gamma)b_h b_v N_v - \delta_v^2\delta_h N_h(\alpha + r_1\gamma + \delta_h)(r_2\gamma + \delta_h)}{\delta_v N_h[\alpha\beta\theta_{hv}(1 - p\gamma)b_h + \delta_v\delta_h(\alpha + r_2\gamma + \delta_h)]}.$$

We denote the above as $\lambda_{vh}^*$. We have the solution

$$E^* = (U_h^*, L_h^*, I_h^*, U_v^*, I_v^*),$$

where

$$U_h^* = \frac{b_h N_h(\alpha + r_1\gamma + \delta_h)(r_2\gamma + \delta_h)}{\delta_h[(\alpha + r_1\gamma + \delta_h)(r_2\gamma + \delta_h) + \lambda_{vh}^*(\alpha + r_2\gamma + \delta_h)]},$$

$$L_h^* = \frac{\lambda_{vh}^* b_h N_h(r_2\gamma + \delta_h)}{\delta_h[(\alpha + r_1\gamma + \delta_h)(r_2\gamma + \delta_h) + \lambda_{vh}^*(\alpha + r_2\gamma + \delta_h)]},$$

$$I_h^* = \frac{\lambda_{vh}^*\alpha b_h N_h}{\delta_h[(\alpha + r_1\gamma + \delta_h)(r_2\gamma + \delta_h) + \lambda_{vh}^*(\alpha + r_2\gamma + \delta_h)]},$$

$$U_v^* = \frac{b_v N_v[\lambda_{vh}^*\delta_h(\alpha + r_2\gamma + \delta_h) + \delta_h(\alpha + r_1\gamma + \delta_h)(r_2\gamma + \delta_h)]}{\lambda_{vh}^*[\alpha\beta\theta_{hv}(1 - p\gamma)b_h + \delta_h\delta_v(\alpha + r_2\gamma + \delta_h)] + \delta_v\delta_h(\alpha + r_1\gamma + \delta_h)(r_2\gamma + \delta_h)}$$

and $\quad$ $$I_v^* = \frac{\lambda_{vh}^*\alpha\beta\theta_{hv}(1 - p\gamma)b_h b_v N_v}{\lambda_{vh}^*\delta_v[\alpha\beta\theta_{hv}(1 - p\gamma)b_h + \delta_h\delta_v(\alpha + r_2\gamma + \delta_h)] + \delta_v^2\delta_h(\alpha + r_1\gamma + \delta_h)(r_2\gamma + \delta_h)}.$$

Note that $E^*$ is biologically feasible if $\lambda_{vh}^* > 0$. Our assumptions on the positivity of parameter values imply that $B$ is always positive. It can also be shown that $A > 0$, i.e.

$$\frac{\alpha\beta^2\theta_{vh}\theta_{hv}(1 - p\gamma)b_h b_v N_v}{\delta_v^2\delta_h N_h(\alpha + r_1\gamma + \delta_h)(r_2\gamma + \delta_h)} > 1.$$

Hence, $U_h^*, L_h^*, I_h^*, U_v^*, I_v^*$ are all positive.

# Appendix B. Computation of the basic reproduction number

Following van Driessche & Watmough [75], we first rearrange our system such that the first $n$ entries contain the infected populations. We have $\dot{X} = (\dot{L}_h, \dot{I}_h, \dot{I}_v, \dot{U}_h, \dot{U}_v)^T$ with solution $X = (L_h, I_h, I_v, U_h, U_v)^T$.

Since there are three compartments for the infected population in our model, we have $n = 3$. Now, $\dot{X}$ can be expressed as

$$\dot{X} = \mathscr{F}(X) - \mathscr{V}(X),$$

where $\mathscr{F}_i(X)$ is the rate of appearance of new infections in compartment $i$, and $\mathscr{V}_i(X) = \mathscr{V}_i^-(X) - \mathscr{V}_i^+(X)$ is the rate of transfer of individuals into compartment $i$ by all other means ($\mathscr{V}_i^-(X)$) and out of compartment $i$ ($\mathscr{V}_i^+(X)$), for $i = 1, \ldots, 5$. In our model, we have

$$\mathscr{F} = \begin{bmatrix} \beta\theta_{\mathrm{vh}}\frac{I_v}{N_h}U_h \\ 0 \\ \beta\theta_{\mathrm{hv}}(1-p\gamma)\frac{I_h}{N_h}U_v \\ 0 \\ 0 \end{bmatrix} \text{ and } \mathscr{V} = \begin{bmatrix} (\alpha + r_1\gamma + \delta_h)L_h \\ -\alpha L_h + (r_2\gamma + \delta_h)I_h \\ \delta_v I_v \\ -b_hN_h - (r_1L_h + r_2I_h)\gamma + \beta\theta_{\mathrm{vh}}\frac{I_v}{N_h}U_h + \delta_hU_h \\ -b_vN_v + \beta\theta_{\mathrm{hv}}(1-p\gamma)\frac{I_h}{N_h}U_v + \delta_vU_v \end{bmatrix}.$$

From here, we obtain the matrices

$$F = \left[\frac{\partial\mathscr{F}_i}{\partial X_j}(X_0)\right] \quad \text{and} \quad V = \left[\frac{\partial\mathscr{V}_i}{\partial X_j}(X_0)\right],$$

where $X_0$ is a disease-free equilibrium and $i, j \in \{1, 2, 3\}$. Diekmann $et\ al.$ [68] defined $\mathcal{R}_0$ as the spectral radius, or the dominant eigenvalue, of the matrix $FV^{-1}$. Following this, we have

$$F(X) = \begin{bmatrix} 0 & 0 & \beta\theta_{\mathrm{vh}}\frac{U_h}{N_h} \\ 0 & 0 & 0 \\ 0 & \beta\theta_{\mathrm{hv}}(1-p\gamma)\frac{U_v}{N_h} & 0 \end{bmatrix},$$

and evaluated at the DFE,

$$F(E_0) = \begin{bmatrix} 0 & 0 & \beta\theta_{\mathrm{vh}}\frac{b_h}{\delta_h} \\ 0 & 0 & 0 \\ 0 & \beta\theta_{\mathrm{hv}}(1-p\gamma)\frac{b_vN_v}{\delta_vN_h} & 0 \end{bmatrix}.$$

We also obtain

$$V(E_0) = \begin{bmatrix} \alpha + r_1\gamma + \delta_h & 0 & 0 \\ -\alpha & r_2\gamma + \delta_h & 0 \\ 0 & 0 & \delta_v \end{bmatrix}$$

and

$$V^{-1}(E_0) = \begin{bmatrix} \frac{1}{\alpha + r_1\gamma + \delta_h} & 0 & 0 \\ \frac{\alpha}{(\alpha + r_1\gamma + \delta_h)(r_2\gamma + \delta_h)} & \frac{1}{r_2\gamma + \delta_h} & 0 \\ 0 & 0 & \frac{1}{\delta_v} \end{bmatrix}.$$

Now, solving for the product of $F(E_0)$ and $V^{-1}(E_0)$, we have

$$F(E_0)V^{-1}(E_0) = \begin{bmatrix} 0 & 0 & \frac{\beta\theta_{\mathrm{vh}}b_h}{\delta_h\delta_v} \\ 0 & 0 & 0 \\ \frac{\alpha\beta\theta_{\mathrm{hv}}(1-p\gamma)b_vN_v}{\delta_vN_h(\alpha + r_1\gamma + \delta_h)(r_2\gamma + \delta_h)} & \frac{\beta\theta_{\mathrm{hv}}(1-p\gamma)b_vN_v}{\delta_vN_h(r_2\gamma + \delta_h)} & 0 \end{bmatrix}.$$

Taking the largest eigenvalue of the above, we have

$$\mathcal{R}_0 = \sqrt{\frac{\alpha\beta^2\theta_{\mathrm{vh}}\theta_{\mathrm{hv}}(1-p\gamma)b_hb_vN_v}{\delta_v^2\delta_hN_h(\alpha + r_1\gamma + \delta_h)(r_2\gamma + \delta_h)}}.$$

# Appendix C. Stability analysis of the steady-state solutions

To study the stability of our equilibrium solutions, we first linearize the system (2.3) near the steady states and calculate the corresponding Jacobian.

At the DFE $E_0 = (b_h N_h/\delta_h, 0, 0, b_v N_v/\delta_v, 0)$, we have

$$
J_{E_0} = \begin{bmatrix}
-\delta_h & r_1\gamma & r_2\gamma & 0 & -\frac{\beta\theta_{vh}b_h}{\delta_h} \\
0 & -(\alpha + r_1\gamma + \delta_h) & 0 & 0 & \frac{\beta\theta_{vh}b_h}{\delta_h} \\
0 & \alpha & -(r_2\gamma + \delta_h) & 0 & 0 \\
0 & 0 & -\frac{\beta\theta_{hv}(1-p\gamma)b_v N_v}{\delta_v N_h} & -\delta_v & 0 \\
0 & 0 & \frac{\beta\theta_{hv}(1-p\gamma)b_v N_v}{\delta_v N_h} & 0 & -\delta_v
\end{bmatrix}.
\tag{C 1}
$$

The corresponding characteristic equation is

$$
\begin{aligned}
0 &= |\lambda I - J_{E_0}| \\
&= (\lambda + \delta_h)(\lambda + \delta_v)\left[(\lambda + \alpha + r_1\gamma + \delta_h)(\lambda + r_2\gamma + \delta_h)(\lambda + \delta_v) - \frac{\alpha\beta^2\theta_{vh}\theta_{hv}(1-p\gamma)b_h b_v N_v}{\delta_v\delta_h N_h}\right].
\end{aligned}
$$

To prove that $E_0$ is locally asymptotically stable, we need to show that the eigenvalues of (C1), i.e. the roots of the characteristic equation above, are all negative. By assumption, $\delta_h$, $\delta_v$ are strictly positive, thus we only need to show that the roots of the cubic polynomial

$$
(\lambda + \alpha + r_1\gamma + \delta_h)(\lambda + r_2\gamma + \delta_h)(\lambda + \delta_v) - \frac{\alpha\beta^2\theta_{vh}\theta_{hv}(1-p\gamma)b_h b_v N_v}{\delta_v\delta_h N_h}
\tag{C 2}
$$

are negative. We have

$$
\begin{aligned}
&(\lambda + \alpha + r_1\gamma + \delta_h)(\lambda + r_2\gamma + \delta_h)(\lambda + \delta_v) - \frac{\alpha\beta^2\theta_{vh}\theta_{hv}(1-p\gamma)b_h b_v N_v}{\delta_v\delta_h N_h} \\
&= \lambda^3 + (\alpha + r_1\gamma + \delta_h + r_2\gamma + \delta_h + \delta_v)\lambda^2 + [(\alpha + r_1\gamma + \delta_h)(r_2\gamma + \delta_h) \\
&\quad + \delta_v(\alpha + r_1\gamma + \delta_h + r_2\gamma + \delta_h)]\lambda + \delta_v(\alpha + r_1\gamma + \delta_h)(r_2\gamma + \delta_h) - \frac{\alpha\beta^2\theta_{vh}\theta_{hv}(1-p\gamma)b_h b_v N_v}{\delta_v\delta_h N_h} \\
&=: a_0\lambda^3 + a_1\lambda^2 + a_2\lambda + a_3.
\end{aligned}
$$

By the Routh–Hurwitz criterion for stability [76], the roots of equation (C 2) are negative if the coefficients of the cubic polynomial above are all positive and $a_1 a_2 > a_0 a_3$. It follows from our assumption on parameter values that

$$
a_0 = 1 > 0, \quad a_1 = \alpha + r_1\gamma + \delta_h + r_2\gamma + \delta_h + \delta_v > 0 \quad \text{and}
$$

$$
a_2 = (\alpha + r_1\gamma + \delta_h)(r_2\gamma + \delta_h) + \delta_v(\alpha + r_1\gamma + \delta_h + r_2\gamma + \delta_h) > 0.
$$

Note that $a_3 > 0$ if and only if $\mathcal{R}_0 < 1$. Indeed,

$$
\begin{aligned}
&\delta_v(\alpha + r_1\gamma + \delta_h)(r_2\gamma + \delta_h) - \frac{b_h b_v N_v \alpha\beta^2\theta_{vh}\theta_{hv}(1-p\gamma)}{\delta_v\delta_h N_h} > 0 \\
&1 - \frac{b_h b_v N_v \alpha\beta^2\theta_{vh}\theta_{hv}(1-p\gamma)}{\delta_v^2\delta_h N_h(\alpha + r_1\gamma + \delta_h)(r_2\gamma + \delta_h)} > 0 \\
&1 - \mathcal{R}_0^2 > 0 \\
&\mathcal{R}_0 < 1.
\end{aligned}
$$

Now, to show that $a_1 a_2 > a_0 a_3$, let

$$
A_1 = \alpha + r_1\gamma + \delta_h, \quad B_1 = r_2\gamma + \delta_h, \quad C_1 = \delta_v, \quad D_1 = \frac{b_h b_v N_v \alpha\beta^2\theta_{vh}\theta_{hv}(1-p\gamma)}{\delta_v\delta_h N_h}.
$$

Then, we can write

$$
a_1 = A_1 + B_1 + C_1, \quad a_2 = A_1 B_1 + C_1(A_1 + B_1), \quad a_3 = A_1 B_1 C_1 - D_1.
$$

Thus,

$$a_1 a_2 - a_0 a_3 = (A_1 + B_1 + C_1)[A_1 B_1 + C_1(A_1 + B_1)] - (A_1 B_1 C_1 - D_1)$$
$$= (A_1 + B_1)[A_1 B_1 + C_1(A_1 + B_1 + C_1)] + D_1$$
$$> 0$$

since $A_1$, $B_1$, $C_1 > 0$ and $D_1 \geq 0$. Thus, the DFE is locally asymptotically stable when $\mathcal{R}_0 < 1$. On the other hand, when $\mathcal{R}_0 > 1$, the coefficient $a_3 < 0$. By Descartes' rule of signs [76], this implies that there will be one positive eigenvalue which proves the instability of $E_0$. Therefore, the disease-free steady state $E_0$ is locally asymptotically stable when $\mathcal{R}_0 < 1$ and unstable when $\mathcal{R}_0 > 1$.

To study the stability of $E^*$, we look at the eigenvalues of the Jacobian matrix evaluated at $E^*$, that is,

$$J_{E^*} = \begin{bmatrix} -(\lambda^*_{vh} + \delta_h) & r_1\gamma & r_2\gamma & 0 & -\beta\theta_{vh}\frac{U^*_h}{N_h} \\ \lambda^*_{vh} & -(\alpha + r_1\gamma + \delta_h) & 0 & 0 & \beta\theta_{vh}\frac{U^*_h}{N_h} \\ 0 & \alpha & -(r_2\gamma + \delta_h) & 0 & 0 \\ 0 & 0 & -\beta\theta_{hv}(1 - p\gamma)\frac{U^*_v}{N_h} & -(\lambda^*_{hv} + \delta_v) & 0 \\ 0 & 0 & \beta\theta_{hv}(1 - p\gamma)\frac{U^*_v}{N_h} & \lambda^*_{hv} & -\delta_v \end{bmatrix}.$$

The corresponding characteristic equation is

$$0 = |\lambda I - J_{E^*}|$$

$$= \begin{vmatrix} \lambda + \lambda^*_{vh} + \delta_h & -r_1\gamma & -r_2\gamma & 0 & \beta\theta_{vh}\frac{U^*_h}{N_h} \\ -\lambda^*_{vh} & \lambda + \alpha + r_1\gamma + \delta_h & 0 & 0 & -\beta\theta_{vh}\frac{U^*_h}{N_h} \\ 0 & -\alpha & \lambda + r_2\gamma + \delta_h & 0 & 0 \\ 0 & 0 & \beta\theta_{hv}(1 - p\gamma)\frac{U^*_v}{N_h} & \lambda + \lambda^*_{hv} + \delta_v & 0 \\ 0 & 0 & -\beta\theta_{hv}(1 - p\gamma)\frac{U^*_v}{N_h} & -\lambda^*_{hv} & \lambda + \delta_v \end{vmatrix}$$

$$= (\lambda + \delta_h)(\lambda + \delta_v)\Big\{ (\lambda + \lambda^*_{hv} + \delta_v)[(\lambda + \alpha + r_1\gamma + \delta_h)(\lambda + r_2\gamma + \delta_h) + \lambda^*_{vh}(\lambda + \alpha + r_2\gamma + \delta_h)]$$

$$- \alpha\beta^2\theta_{vh}\theta_{hv}(1 - p\gamma)\frac{U^*_h U^*_v}{N_h^2} \Big\}.$$

Since $\delta_h$, $\delta_v$ are strictly positive for all time $t$, $t \geq 0$, we only need to show that the roots of the cubic polynomial

$$(\lambda + \lambda^*_{hv} + \delta_v)\Big[(\lambda + \alpha + r_1\gamma + \delta_h)(\lambda + r_2\gamma + \delta_h) + \lambda^*_{vh}(\lambda + \alpha + r_2\gamma + \delta_h)\Big] - \alpha\beta^2\theta_{vh}\theta_{hv}(1 - p\gamma)\frac{U^*_h U^*_v}{N_h^2}$$

(C 3)

are negative. Substituting the values for $U^*_h$ and $U^*_v$, we obtain

$$(\lambda + \lambda^*_{hv} + \delta_v)\Big[(\lambda + \alpha + r_1\gamma + \delta_h)(\lambda + r_2\gamma + \delta_h) + \lambda^*_{vh}(\lambda + \alpha + r_2\gamma + \delta_h)\Big] - \alpha\beta^2\theta_{vh}\theta_{hv}(1 - p\gamma)\frac{U^*_h U^*_v}{N_h^2}$$

$$= \lambda^3 + (\lambda^*_{hv} + \delta_v + \lambda^*_{vh} + \alpha + r_1\gamma + \delta_h + r_2\gamma + \delta_h)\lambda^2 + \Big[(\alpha + r_1\gamma + \delta_h)(r_2\gamma + \delta_h) + \lambda^*_{vh}(\alpha + r_2\gamma + \delta_h)$$

$$+ (\lambda^*_{hv} + \delta_v)(\lambda^*_{vh} + \alpha + r_1\gamma + \delta_h + r_2\gamma + \delta_h)\Big]\lambda$$

$$+ (\lambda^*_{hv} + \delta_v)\Big[(\alpha + r_1\gamma + \delta_h)(r_2\gamma + \delta_h) + \lambda^*_{vh}(\alpha + r_2\gamma + \delta_h)\Big]$$

$$- \frac{b_h b_v N_v \alpha\beta^2\theta_{vh}\theta_{hv}(1 - p\gamma)(\alpha + r_1\gamma + \delta_h)(r_2\gamma + \delta_h)}{\delta_h N_h(\lambda^*_{hv} + \delta_v)[\lambda^*_{vh}(\alpha + r_2\gamma + \delta_h) + (\alpha + r_1\gamma + \delta_h)(r_2\gamma + \delta_h)]}$$

$$=: b_0\lambda^3 + b_1\lambda^2 + b_2\lambda + b_3.$$

By the Routh–Hurwitz criterion for stability, the roots of (C 3) are negative if the coefficients of the cubic polynomial above are positive and $b_1 b_2 > b_0 b_3$. It follows from our assumption on the parameter values that

$$b_0 = 1 > 0,$$
$$b_1 = (\lambda_{hv}^* + \delta_v + \lambda_{vh}^* + \alpha + r_1\gamma + \delta_h + r_2\gamma + \delta_h) > 0, \text{ and}$$
$$b_2 = (\alpha + r_1\gamma + \delta_h)(r_2\gamma + \delta_h) + \lambda_{vh}^*(\alpha + r_2\gamma + \delta_h) + (\lambda_{hv}^* + \delta_v)(\lambda_{vh}^* + \alpha + r_1\gamma + \delta_h + r_2\gamma + \delta_h) > 0.$$

Note that $b_3 > 0$ if and only if

$$(\lambda_{hv}^* + \delta_v)\Big[(\alpha + r_1\gamma + \delta_h)(r_2\gamma + \delta_h) + \lambda_{vh}^*(\alpha + r_2\gamma + \mu_h)\Big]$$
$$- \frac{\alpha\beta^2\theta_{vh}\theta_{hv}(1 - p\gamma)(\alpha + r_1\gamma + \delta_h)(r_2\gamma + \delta_h)b_h b_v N_v}{\delta_h N_h(\lambda_{hv}^* + \delta_v)[\lambda_{vh}^*(\alpha + r_2\gamma + \delta_h) + (\alpha + r_1\gamma + \delta_h)(r_2\gamma + \delta_h)]} > 0.$$

Following a rigorous computation, the above inequality is equivalent to $\mathcal{R}_0^2 > \mathcal{R}_0$ which means $\mathcal{R}_0 > 1$ since $\mathcal{R}_0 > 0$. Now, to show that $b_1 b_2 > b_0 b_3$, let

$$A_2 = \alpha + r_1\gamma + \delta_h, \quad B_2 = r_2\gamma + \delta_h,$$
$$C_2 = \alpha + r_2\gamma + \delta_h, \quad D_2 = \lambda_{hv}^* + \delta_v,$$

and 
$$E_2 = \frac{b_h b_v N_v \alpha\beta^2\theta_{vh}\theta_{hv}(1 - p\gamma)(\alpha + r_1\gamma + \delta_h)(r_2\gamma + \delta_h)}{\delta_h N_h(\lambda_{hv}^* + \delta_v)[\lambda_{vh}^*(\alpha + r_2\gamma + \delta_h) + (\alpha + r_1\gamma + \delta_h)(r_2\gamma + \delta_h)]}.$$

Then, we can write

$$b_1 = \lambda_{vh}^* + A_2 + B_2 + D_2, \quad b_2 = A_2 B_2 + \lambda_{vh}^* C_2 + \Big(\lambda_{vh}^* + A_2 + B_2\Big)D_2 \quad \text{and}$$

$$b_3 = D_2\Big(A_2 B_2 + \lambda_{vh}^* C_2\Big) - E_2.$$

We have

$$b_1 b_2 - b_0 b_3 = (\lambda_{vh}^* + A_2 + B_2)[A_2 B_2 + \lambda_{vh}^* C_2 + \Big(\lambda_{vh}^* + A_2 + B_2\Big)D_2]$$
$$+ D_2\Big(A_2 B_2 + \lambda_{vh}^* C_2\Big) + \Big(\lambda_{vh}^* + A_2 + B_2\Big)D_2^2 - D_2\Big(A_2 B_2 + \lambda_{vh}^* C_2\Big) + E_2$$
$$> 0$$

since $\lambda_{vh}^* > 0$, $A_2$, $B_2$, $C_2$, $D_2 > 0$ and $E_2 \geq 0$. Hence, the endemic equilibrium $E^*$ is locally asymptotically stable when $\mathcal{R}_0 > 1$.

# Appendix D. Proof of the existence of the optimal control $u^*(t)$

The existence of the optimal control $u^*(t)$ such that $J(u^*(t)) = \min_\Omega u(t)$ with state system

$$\frac{dU_h}{dt} = b_h N_h + (r_1 L_h + r_2 I_h)u(t) - \beta\theta_{vh}\frac{I_v}{N_h}U_h - \delta_h U_h,$$

$$\frac{dL_h}{dt} = \beta\theta_{vh}\frac{I_v}{N_h}U_h - (\alpha + r_1 u(t) + \delta_h)L_h,$$

$$\frac{dI_h}{dt} = \alpha L_h - (r_2 u(t) + \delta_h)I_h,$$

$$\frac{dU_v}{dt} = b_v N_v - \beta\theta_{hv}(1 - pu(t))\frac{I_h}{N_h}U_v - \delta_v U_v$$

and 
$$\frac{dI_v}{dt} = \beta\theta_{hv}(1 - pu(t))\frac{I_h}{N_h}U_v - \delta_v I_v,$$

is given by the convexity of the objective functional integrand. The adjoint equations and transversality conditions are obtained using Pontryagin's maximum principle [72]. In particular, differentiation of the

Hamiltonian $H$ with respect to the state variables gives the following system:

$$\frac{d\lambda_1}{dt} = -\frac{\partial H}{\partial U_h}, \quad \frac{d\lambda_2}{dt} = -\frac{\partial H}{\partial L_h}, \quad \frac{d\lambda_3}{dt} = -\frac{\partial H}{\partial I_h}, \quad \frac{d\lambda_4}{dt} = -\frac{\partial H}{\partial U_v}, \quad \frac{d\lambda_5}{dt} = -\frac{\partial H}{\partial I_v}$$

with $\lambda_i(t_f) = 0$, for $i = 1, \ldots, 5$.

The optimal control $u^*(t)$ is derived using the following optimality condition:

$$\frac{\partial H}{\partial u} = cu + r_1 L_h(\lambda_1 - \lambda_2) + r_2 I_h(\lambda_1 - \lambda_3) + \beta\theta_{\text{hv}} p \frac{I_h U_v}{N_h}(\lambda_4 - \lambda_5) = 0$$

at $u = u^*(t)$ on the set $\Omega$. Hence, we have

$$u^*(t) = \frac{1}{c}\left[ r_1 L_h(\lambda_2 - \lambda_1) + r_2 I_h(\lambda_3 - \lambda_1) + \beta\theta_{\text{hv}} p \frac{I_h U_v}{N_h}(\lambda_5 - \lambda_4) \right].$$

Taking into account the bounds on the control, we obtain the following characterization of $u^*(t)$:

$$u^*(t) = \min\left( u_{\max}, \left( u_{\min}, \frac{1}{c}\left[ r_1 L_h(\lambda_2 - \lambda_1) + r_2 I_h(\lambda_3 - \lambda_1) + \beta\theta_{\text{hv}} p \frac{I_h U_v}{N_h}(\lambda_5 - \lambda_4) \right] \right) \right).$$

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
