## [Peer Review File · Royal Society Open Science]

Review History

RSOS-201965.R0 (Original submission)

Review form: Reviewer 1

Is the manuscript scientifically sound in its present form?

Yes

Are the interpretations and conclusions justified by the results?

Yes

Is the language acceptable?

Yes

Do you have any ethical concerns with this paper?

No

Have you any concerns about statistical analyses in this paper?

No

Recommendation?

Accept with minor revision (please list in comments)

Comments to the Author(s)

There are several instance where the authors use the expressions ...in this paper, in this work, in this section. They can minimize this. For example, "In this study, minimizing the infected mosquito ..." can be replaced by "Herein, we minimize the infected mosquito ..."

Review form: Reviewer 2

Is the manuscript scientifically sound in its present form?

Yes

Are the interpretations and conclusions justified by the results?

No

Is the language acceptable?

Yes

Do you have any ethical concerns with this paper?

No

Have you any concerns about statistical analyses in this paper?

No

Recommendation?

Major revision is needed (please make suggestions in comments)

Comments to the Author(s)

The result section needs to be re-written with more emphasis on the percent improvement of the vaccine application for each scenarios. Emphasis should be made to show how is the model different from previously existing in literature.

Decision letter (RSOS-201965.R0)

Dear Ms Salonga

On behalf of the Editors, we are pleased to inform you that your Manuscript RSOS-201965 "A mathematical model of the dynamics of lymphatic filariasis in Caraga Region, the Philippines" has been accepted for publication in Royal Society Open Science subject to minor revision in accordance with the referees' reports. Please find the referees' comments along with any feedback from the Editors below my signature.

Please submit your revised manuscript and required files (see below) no later than 7 days from today's (ie 07-Apr-2021) date. Note: the ScholarOne system will 'lock' if submission of the revision is attempted 7 or more days after the deadline. If you do not think you will be able to meet this deadline please contact the editorial office immediately.

on behalf of Professor Joshua Ross (Associate Editor) and Mark Chaplain (Subject Editor)
openscience@royalsociety.org

Reviewer comments to Author:
Reviewer: 1

Comments to the Author(s)
There are several instance where the authors use the expressions ...in this paper, in this work, in this section. They can minimize this. For example, "In this study, minimizing the infected mosquito ..." can be replaced by "Herein, we minimize the infected mosquito ..."

Reviewer: 2

Comments to the Author(s)
The result section needs to be re-written with more emphasis on the percent improvement of the vaccine application for each scenarios. Emphasis should be made to show how is the model different from previously existing in literature.

===PREPARING YOUR MANUSCRIPT===

Your revised paper should include the changes requested by the referees and Editors of your manuscript. You should provide two versions of this manuscript and both versions must be provided in an editable format:
one version identifying all the changes that have been made (for instance, in coloured highlight, in bold text, or tracked changes);
a 'clean' version of the new manuscript that incorporates the changes made, but does not highlight them. This version will be used for typesetting.

===PREPARING YOUR REVISION IN SCHOLARONE===

- If you are providing image files for potential cover images, please upload these at this step, and inform the editorial office you have done so. You must hold the copyright to any image provided.
- A copy of your point-by-point response to referees and Editors. This will expedite the preparation of your proof.

- Ensure that your data access statement meets the requirements at <https://royalsociety.org/journals/authors/author-guidelines/#data>. You should ensure that you cite the dataset in your reference list. If you have deposited data etc in the Dryad repository, please only include the 'For publication' link at this stage. You should remove the 'For review' link.
- If you are requesting an article processing charge waiver, you must select the relevant waiver option (if requesting a discretionary waiver, the form should have been uploaded at Step 3 'File upload' above).
- If you have uploaded ESM files, please ensure you follow the guidance at <https://royalsociety.org/journals/authors/author-guidelines/#supplementary-material> to include a suitable title and informative caption. An example of appropriate titling and captioning may be found at https://figshare.com/articles/Table_S2_from_Is_there_a_trade-off_between_peak_performance_and_performance_breadth_across_temperatures_for_aerobic_scope_in_teleost_fishes_/3843624.

Author's Response to Decision Letter for (RSOS-201965.R0)

See Appendix A.

RSOS-201965.R1 (Revision)

Review form: Reviewer 2

Is the manuscript scientifically sound in its present form?

Yes

Are the interpretations and conclusions justified by the results?

Yes

Is the language acceptable?

Yes

Do you have any ethical concerns with this paper?

Yes

Have you any concerns about statistical analyses in this paper?

Yes

Recommendation?

Accept as is

Comments to the Author(s)

The paper has been revised and changes for the better has been made.

Decision letter (RSOS-201965.R1)

Dear Ms Salonga,

It is a pleasure to accept your manuscript entitled "A mathematical model of the dynamics of lymphatic filariasis in Caraga Region, the Philippines" in its current form for publication in Royal Society Open Science. The comments of the reviewer(s) who reviewed your manuscript are included at the foot of this letter.

on behalf of Professor Joshua Ross (Associate Editor) and Mark Chaplain (Subject Editor)
openscience@royalsociety.org

Reviewer comments to Author:

Reviewer: 2

Comments to the Author(s)

The paper has been revised and changes for the better has been made.

Appendix A

13 April 2021

Prof. Mark Chaplain
Mathematics Subject Editor
Royal Society Open Science

Dear Prof. Chaplain;

Greetings!

We would like to submit our revised article (ID: RSOS-201965) titled "*A mathematical model of the dynamics of lymphatic filariasis in Caraga Region, the Philippines*" for possible publication in Royal Society Open Science.

Thank you for seeing the merits of our work. We appreciate all the valuable comments and recommendations of the reviewers. We modified the manuscript accordingly. Our detailed response to the reviewers can be found below.

We also made some changes to make the presentation of the paper clearer. These changes are written in blue text (see lines 215-217 and the caption in Table 3). The text in lines 238-245, which were previously in Section 3c, were transferred to Section 2b since these are information on the data. Furthermore, the manuscript is formatted according to the template provided.

If you have questions regarding this submission, please do not hesitate to contact me using the below-mentioned email address.

We are hoping for a favorable response. Thank you very much.

Respectfully,

Pamela Kim N. Salonga
Corresponding Author
Institute of Mathematics
University of the Philippines Diliman
pksalonga@math.upd.edu.ph

Response to Reviewer 1

The changes in the manuscript addressing the comments of Reviewer 1 are colored magenta.

Comment	Response
There are several instance where the authors use the expressions ...in this paper, in this work, in this section. They can minimize this. For example, "In this study, minimizing the infected mosquito ..." can be replaced by "Herein, we minimize the infected mosquito ..."	Thank you for your suggestion. We scanned the whole manuscript and modified the expressions containing "in this paper, in this work, in this section". Please see lines 144, 174, 234, 260-261, 339-340, and 409.

Response to Reviewer 2

The changes in the manuscript addressing the comments of Reviewer 2 are colored red.

Comments	Response
The result section needs to be re-written with more emphasis on the percent improvement of the vaccine application for each scenarios.	Your comments have greatly improved the presentation of our results. To emphasize the improvement of the vaccine application, we added a table that shows the percent reduction in the infectious population when the optimal control using different implementation costs is applied. Please see Table 4 on page 17. The discussion of these results can be found in lines 449-451, 455-458, and 461-462. The caption of Figure 6 is also modified. Furthermore, we modified parts of the manuscript to highlight the impact of mass drug administration (MDA) in controlling the spread of lymphatic filariasis (LF), reflected by the results obtained in the sensitivity analysis and parameter estimation. Please see lines 207, 384-390, 417-420, 476-478, and 480-481.
Emphasis should be made to show how is the model different from previously existing in literature.	We appreciate this suggestion. To the best of our knowledge, our study provides the first mathematical model of LF in the Philippines. Compared to existing models, we consider a more realistic representation of MDA wherein the antifilarial drugs are given to all eligible individuals in the population, infected and uninfected alike. In the existing models of LF found in the literature, the parameter values are either assumed or obtained from existing malaria models. In our paper, we use Philippine filariasis data to estimate model parameters to have more meaningful insights. We used current techniques (e.g. sensitivity analysis using LHS and PRCC, bootstrapping, and uncertainty analysis) to quantify uncertainty in the obtained parameters estimates. We modified the manuscript to emphasize these contributions and to highlight the difference of our model from the existing models in the literature. Kindly refer to lines 91-98, and 112-122.